# PML Regulates the Epidermal Differentiation Complex and Skin Morphogenesis during Mouse Embryogenesis

**DOI:** 10.3390/genes11101130

**Published:** 2020-09-25

**Authors:** Anna Połeć, Alexander D. Rowe, Pernille Blicher, Rajikala Suganthan, Magnar Bjørås, Stig Ove Bøe

**Affiliations:** 1Department of Medical Biochemistry, Oslo University Hospital, 0372 Oslo, Norway; Anna.Polec@rr-research.no; 2Department of Medical Biochemistry, Institute for Clinical Medicine, University of Oslo, 0372 Oslo, Norway; alexarow@ifi.uio.no (A.D.R.); pernille.blicher@medisin.uio.no (P.B.); 3Department of Newborn Screening, Division of Child and Adolescent Medicine, Oslo University Hospital, 0372 Oslo, Norway; 4Department of Microbiology, Oslo University Hospital, 0373 Oslo, Norway; Rajikala.Suganthan@rr-research.no (R.S.); magnar.bjoras@ntnu.no (M.B.); 5Department of Clinical and Molecular Medicine, Norwegian University of Science and Technology (NTNU), 7028 Trondheim, Norway

**Keywords:** PML, promyelocytic leukemia, EDC/epidermal differentiation complex, epidermis, keratinocytes, skin, mouse embryo, development

## Abstract

The promyelocytic leukemia (PML) protein is an essential component of nuclear compartments called PML bodies. This protein participates in several cellular processes, including growth control, senescence, apoptosis, and differentiation. Previous studies have suggested that PML regulates gene expression at a subset of loci through a function in chromatin remodeling. Here we have studied global gene expression patterns in mouse embryonic skin derived from Pml depleted and wild type mouse embryos. Differential gene expression analysis at different developmental stages revealed a key role of PML in regulating genes involved in epidermal stratification. In particular, we observed dysregulation of the late cornified envelope gene cluster, which is a sub-region of the epidermal differentiation complex. In agreement with these data, PML body numbers are elevated in basal keratinocytes during embryogenesis, and we observed reduced epidermal thickness and defective hair follicle development in PML depleted mouse embryos.

## 1. Introduction

The promyelocytic leukemia (PML) protein has been implicated in various cellular functions including apoptosis [1], genome maintenance [2,3,4], innate immunity [5,6], and differentiation [7,8,9]. A unique feature of PML is its ability to form nuclear compartments called PML bodies [10]. These bodies typically range in size from 0.1 to 1.0 µm in diameter, and their number per nucleus commonly varies between 4 and 30. Although PML is the only protein that so far has been shown to be required for PML body assembly, a large number of additional proteins (more than 150) have been identified as PML body constituents [11,12,13]. A large number of these PML body resident proteins participate in gene expression through their role as transcription factors (e.g., STAT3, SP1, and GATA2), transcription factor modifying enzymes (e.g., PIAS1, SENP1, and HIPK2), or chromatin modifiers (e.g., DAXX, CBP, and HDACs). In agreement with this, PML bodies are known to make contacts with specific chromatin regions that frequently are associated with transcriptionally active genes [14,15,16].

A well characterized example of PML mediated gene expression regulation comes from studies showing that PML interacts with the major histocompatibility complex (MHC) locus. This gene cluster, which comprises functionally related genes essential for the acquired immune system to recognize foreign molecules, is regulated through chromatin remodeling [17,18]. PML bodies have been shown to interact with the MHC gene locus [19,20], and in a study by Ulbricht et al. this interaction was found to be controlled by interferon-mediated gene regulation [20]. In addition, PML was found to regulate transcription of genes from the MHCII cluster through interaction with class II transactivator (CIITA) and the MHC class I locus through interaction with the chromatin loop organizer protein SATB1 [21].

Several previous studies have revealed a key role of PML in development and morphogenesis of various tissues and organs. For example, PML has been shown to influence the process of hematopoiesis through maintenance of hematopoietic stem cells [22,23]. In addition, this protein has been implicated in differentiation and/or regeneration of brain [7,8], mammary gland [9], bone [24], and muscle [25]. Lastly, PML has been shown to regulate the balance of self-renewal and differentiation in pluripotent stem cells [26].

In the present study we have investigated the role of PML in the development of epidermis. By employing RNA-seq analysis of wild type and PML depleted mice at different stages of embryonic development, we identified the epidermal differentiation complex (EDC), a gene cluster with striking structural similarities to the MHC, as one of the most deregulated loci. It comprises 62 coding genes within four gene families: filaggrin and FLG-like, late cornified envelope genes (LCEs), small proline-rich region (SPRRs, including loricrin (LOR) and involucrin (IVL)), and S100 genes [27,28]. The genes within this cluster share a common function in terminal differentiation of epidermal cells, and PML is found to primarily regulate the LCE genes during the last stages of embryonal development, which involves epidermal stratification. In agreement with a role of PML in epidermal stratification and morphogenesis we show that PML knockout mice exhibit a delayed epidermal stratification and impaired hair follicle development during embryogenesis. The study demonstrates a role of PML in epidermis development during embryogenesis and points to an important role of PML in gene cluster regulation.

## 2. Materials and Methods

### 2.1. Mice

129 *Pml*−/− mice were obtained from National Cancer Institute Repository in Frederick, MD, USA [29]. Mice were backcrossed to C57BL/6 mice for more than eight generations. C57BL6 was used as wild type control. All animal experiments were performed in accordance with the guidelines of the Oslo University Hospital Rikshospitalet (OUS) and all experimental procedures have been approved by the Norwegian Animal Research Committee (FOTS reference no. 1096).

### 2.2. Antibodies

Primary antibodies: mouse monoclonal anti-PML (Santa Cruz; PG-M3; 1:400), mouse monoclonal anti-PML (Millipore; clone 36.1-104; 1:400), rabbit monoclonal anti-K15 (Abcam; ab52816; 1:400), rabbit anti-K14 (Abcam; ab175549; 1:400), rabbit anti-K5 (Abcam; ab24647; 1:400), and K10 (Abcam; RKSE60; 1:200). Secondary antibodies used: Alexa Fluor^®^ 488 goat anti-mouse IgG (H + L) (Molecular Probes; 1:500) and Alexa Fluor^®^ 594 goat anti-rabbit IgG (H + L) (Molecular Probes; 1:500). 

### 2.3. Isolation and Culturing of Epidermal Keratinocytes

Mouse epidermal keratinocytes were isolated from skin using a procedure adapted from CELLnTEC’s protocol for primary and long-term cell cultures (http://cellntec.com/wp-content/uploads/pdf/General_Cultivation.pdf). Briefly, skin from the dorsal region of stage E18.5 embryos was dissected, while the adipocyte layer and most of the dermis was discarded. The extracted skin tissue was cut into smaller pieces (~0.5 cm × 1 cm in size) that were placed in a 15 mL tube containing 5 mL 1 × dispase II (Merck, Darmstadt, Germany; D4693) and incubated at 4 °C for 15 h. Next, the minced skin was placed in a petri dish in CNT medium, epidermal side up. Subsequently, epidermal layers were peeled off using forceps and then placed in a 15 mL tube with PBS. The epidermal tissue was then subjected to centrifugation for 5 min at 100× *g* and subsequently transferred to a new tube before treatment with 0.5–1 mL trypsin at 37 °C for 4 min. Following vigorous suspension of the tissue (by pipetting up and down), cells were subjected to filtration through a 70 µm Falcon cell strainer (Corning, Corning, NY, USA). Filtered cells were then transferred to a 15 mL tube, where trypsin was neutralized by adding an equal volume of soybean trypsin inhibitor. Cells were then subjected to centrifugation for 5 min at 100× *g* and the resulting cell pellet was suspended in 1 mL CNT medium. 

To culture the isolated skin cells, approximately 5 × 10^6^ cells were seeded in a 100 mm dish coated with collagen IV. Following a recovery period of three days at 37 °C in a humidified CO_2_ incubator, the medium was changed regularly every third day. All experiments on isolated skin cells in culture were performed between passage 1 and 3.

### 2.4. Immunohistochemistry (IHC) Analysis

Embryos and skin samples were fixed and embedded in paraffin as previously described^7^. Samples were sectioned at 4–7 µm on a Rotary Microtome HM355S (Thermo Fisher Scientific, Waltham, MA, USA) and mounted on SuperFrost Plus slides (VWR). Slides were baked at 57–58 °C for 10 min, followed by consecutive washes in Neo-Clear (Merck, Darmstadt, Germany; 2 × 5 min) 100% ethanol (2 × 3 min), 96% ethanol (1 min), and 70% ethanol (1 min) before placing in Milli-Q (MQ) water. Epitope retrieval was carried out by placing the slides in preheated antigen retrieval buffer (10 mM Sodium citrate (Thermo Fisher Scientific, Waltham, MA, USA), 0.05% Tween 20 (Sigma-Aldrich), pH 6.0) in a pressure cooker for 3 min. Subsequently, slides were cooled down in running cold water for 10 min, rinsed with washing buffer, and blocked with 5% goat serum (GS; Merck, Darmstadt, Germany) and 5% bovine serum albumin (BSA) in PBS for 30 min at room temperature (RT). Primary antibodies were diluted in staining buffer (PBS with 0.5% BSA 0.5% GS) and incubated at 4 °C overnight. After rinsing with washing buffer, samples were incubated with secondary antibodies in staining buffer for 2 h at 37 °C. Finally, samples were incubated for 5 min at RT with DAPI (1:1000), rinsed with washing buffer, and mounted using Vectashield mounting medium (Vector Laboratories, Burlingame, CA, USA). Microscopy analysis of immunofluorescently labeled tissue sections was performed using a Leica SP8 confocal microscope equipped with a 40x oil immersion objective.

### 2.5. RNA-Seq Analysis

RNA was isolated from mouse embryos using the All-Prep Kit (Qiagen, Valencia, CA, USA). Briefly, tissue was homogenized in lysis buffer RLT using Fastprep-24 instrument, centrifuged for 3 min at 30,000× *g,* and supernatant was used for total RNA isolation. RNA concentration was measured by NanoDrop Spectrophotometer. Library construction, sequencing, and data processing were performed through a commercially available service by Beijing Genomics Institute (BGI, Hong Kong, China). The filtered data are available in Appendix A.

### 2.6. qPCR Analysis

RNA extraction from embryos was performed using the method described by Chomczynski and Sacchi [30]. Briefly, 30 mg of tissue was lysed in 1 mL Solution D and RNA was separated from DNA and proteins by the addition of 100 mL sodium acetate, 1 mL phenol, and 20 mL chloroform: isoamyl alcohol (24:1) and centrifugation, followed by precipitation with isopropanol.

One microgram of total RNA was then reverse-transcribed using the High-Capacity cDNA Reverse Transcription Kit from Applied Biosystems. Real-time quantitative PCR (RT-qPCR) was performed using the ABI 7900HT system (Applied Biosystems, Waltham, MA, USA) with Power SYBR Green PCR master mix (Life Technologies, Carlsbad, CA, USA) according to the system instructions. Mouse *Pgk1* house-keeping gene was used as reference gene for normalization in all of the experiments. Melting point analysis was performed to confirm the specificity of the PCR products and a standard curve for each primer pair was generated to determine primer efficacy and to confirm that the PCR reaction was run in the linear range. Relative gene expression was calculated using the method by Michael W. Pfaffl [31]. The primers used are depicted in Table 1.

## 3. Results

### 3.1. PML Regulates EDC Genes during Embryogenesis

To analyze a role of PML in gene expression during development of epidermis, we performed genome-wide RNA-seq analysis in epidermal tissue obtained at different stages of embryonic development. Samples were collected at four different stages of embryogenesis: (1) Embryonic day (E) 13.5, a developmental stage where the epidermis generally consists of a single layer of basal keratinocytes, (2) E15.5 and E16.5, which represent early and late stages of epidermal stratification, respectively, and (3) E17.5, which represents fully developed epidermis. Comparison between wild type (wt) and PML depleted embryos revealed a specific up-regulation of genes within the EDC gene locus at E15.5 in knockout mice versus wt, suggesting that PML negatively regulates genes within this locus at certain stages of epidermal development (Figure 1A,B). Upon closer inspection, we noted that PML primarily affected expression of EDC genes corresponding to the LCE cluster, while most genes within the SPRR and S100 region generally remained unaffected (Appendix A).

We next analyzed expression of selected LCE genes at different time points of mouse development using real-time (RT) PCR. In agreement with the genome-wide RNA seq analysis, these data also showed an increase of LCE gene expression in *Pml*−/− mice versus *Pml*+/+ mice at E15.5. Interestingly, these data also revealed PML regulated gene expression at stage E18.5 (a time-point that was not included in the RNA-seq analysis) which generally denotes the last day of mouse embryogenesis (Figure 2A,B).

Finally, we analyzed keratinocytes that had been growing in culture following isolation from epidermis dissected from the dorsal region of stage E18.5 embryos. While one of the selected genes (*Lce1e*) showed similar gene expression in wt and PML KO mouse cells, three other LCE loci tested (*Ivl*, *Lce1b*, and *Lce1h*) exhibited a significant increase in gene expression from the *Pml*−/− cells compared to the *Pml*+/+ cells (Figure 2C). Combined, these experiments reveal a role of PML in temporal regulation of the LCE sub-region within the EDC locus during epidermis development.

### 3.2. PML Expression Is Elevated in Basal Keratinocytes during Embryogenesis

We next investigated the subcellular localization and tissue scale distribution of PML in epidermal tissue. For these experiments we carried out co-immunostaining using antibodies against K15, which is a marker of basal epidermal cells, in combination with antibodies against PML. In the embryonic tissue, we detected PML bodies primarily in basal epidermal cells, while K15 negative suprabasal cells as well as neighboring endothelial cells (directly underneath the epidermal cell layers) were found to contain significantly fewer PML bodies per nucleus (Figure 3A–C). Notably, the number of PML bodies in basal epidermal cells was observed to be significantly higher during epidermal stratification at E13.5, E15.5, and E17.5, compared to fully developed epidermis at E18.5 and postnatal stage (P0) (Figure 3B,C).

### 3.3. PML Depletion Affects Epidermal Thickness and Hair Follicle Size during Embryogenesis

To analyze potential effects of PML depletion on epidermis growth and morphology, we first investigated differences in thickness of basal and suprabasal epidermal layers. We observed significantly thicker suprabasal layers in wt compared to PML depleted tissue samples at stage E17.5 and P0 (Figure 4A,B). In addition, we detected increased cell densities within the suprabasal layers in wt versus PML depleted epidermal tissue (Figure 4C). This observation may indicate that the decrease in suprabasal thickness observed for *Pml*−\− cells is, in part, due to an increase in tissue compactness.

We next analyzed potential differences in hair follicle size and morphology. For these experiments we measured the length and angle of hair follicles relative to the epidermal surface (Figure 5A,B). We found that the size of hair follicles was considerably larger in wt versus PML depleted mouse at E17.5 and P0 (Figure 5C). In addition, we noted that the hair follicle angles relative to the epidermal surface in PML depleted mouse deviated from that observed in wt animals (Figure 5C). These results are in agreement with a role of PML in epidermal development and hair follicle morphogenesis during embryogenesis.

## 4. Discussion

In the present study we have investigated the effect of PML depletion on the development of epidermal tissue in mice during embryogenesis. We have shown that the absence of PML leads to reduced thickness of suprabasal epidermal layers and incompletely developed hair follicles, especially at stages of embryogenesis where stratification is nearing completion. This result is consistent with previous studies showing that PML is required for the development and regeneration of other organs such as brain [7,8], mammary gland [9], bone [24], muscle [25], and the hematopoietic system [22,23].

The mechanism by which PML regulates epidermal development could be linked to transcriptional control of genes involved in cell fate decision and differentiation. This is in agreement with the higher amount of detectable PML bodies in basal keratinocytes compared to more differentiated skin cells in suprabasal layers during development. It is also consistent with previous studies showing that PML regulates stem cell fate and is expressed at higher levels in stem cells compared to differentiated cells.

A role of PML in transcription regulation and development of epidermis is also supported by the genome-wide gene expression analysis. Several of the most differentially expressed genes detected by comparing data from wild type and PML depleted mice at different stages of embryogenesis were found to be present within the EDC, a gene-rich locus which is known to encode multiple proteins with key functions in epidermal growth and differentiation. Previous studies have shown that EDC is regulated at the level of chromatin remodeling by p62, one of the master transcription factors involved in epidermal differentiation and morphogenesis. For example, Fessing et al. demonstrated that p63 regulates EDC remodeling and gene expression through activation of the genome organizer Satb1 [32]. In a subsequent study p63 was shown to orchestrate nuclear re-localization and expression of EDC genes through transcription activation of the chromatin remodeler Brg1 [33]. It should be noted that PML depletion primarily affects only one of the four EDC sub-regions, namely the segment that encodes the LCE proteins. A similar sub-regional effect of PML depletion was also observed for the MHC locus [20,21]. Given the similarities between MHC and EDC (two of the largest gene clusters identified within the mammalian genome), it is conceivable that PML regulates gene clusters within these two elements at the chromatin level by similar mechanisms. Further studies should clarify the mechanism whereby PML regulates EDC in the context of a three dimensional chromatin.

## 5. Conclusions

In the present study we demonstrate a role of PML in regulating a subset of genes within the EDC gene cluster during embryonic development in mouse. In agreement with this, we observed impaired epidermal stratification and hair follicle development in PML depleted mice. Since EDC have structural features in common with another large gene cluster known to be targeted by PML, namely the MHC, it is conceivable that PML regulates the two loci through a common mechanism at the level of chromatin remodeling.

## Figures and Tables

**Figure 1 genes-11-01130-f001:**
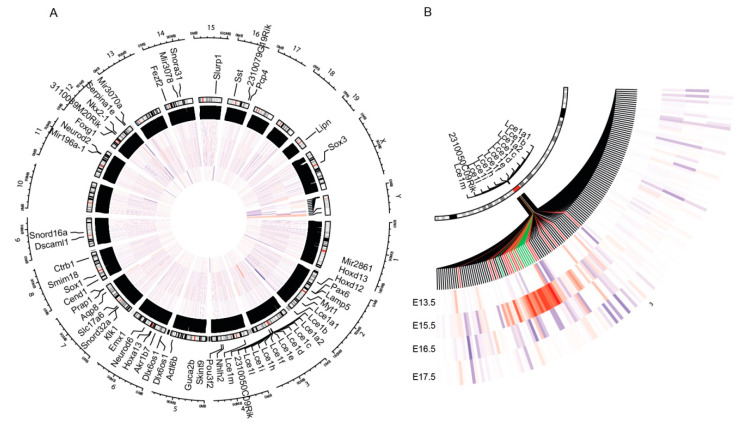
Promyelocytic leukemia (PML) regulates epidermal differentiation complex (EDC) genes during embryogenesis. (**A**) Heat map diagram of genes differentially affected in RNA-seq analysis of extracts from wild type (wt) and PML knockout mouse epidermis. Genome-wide differential expression at stage E13.5 (inner circle), E15.5, E16.5 (middle circles), and E17.5 (outer circle) is depicted. (**B**) Zoom-in of the region containing the EDC locus.

**Figure 2 genes-11-01130-f002:**
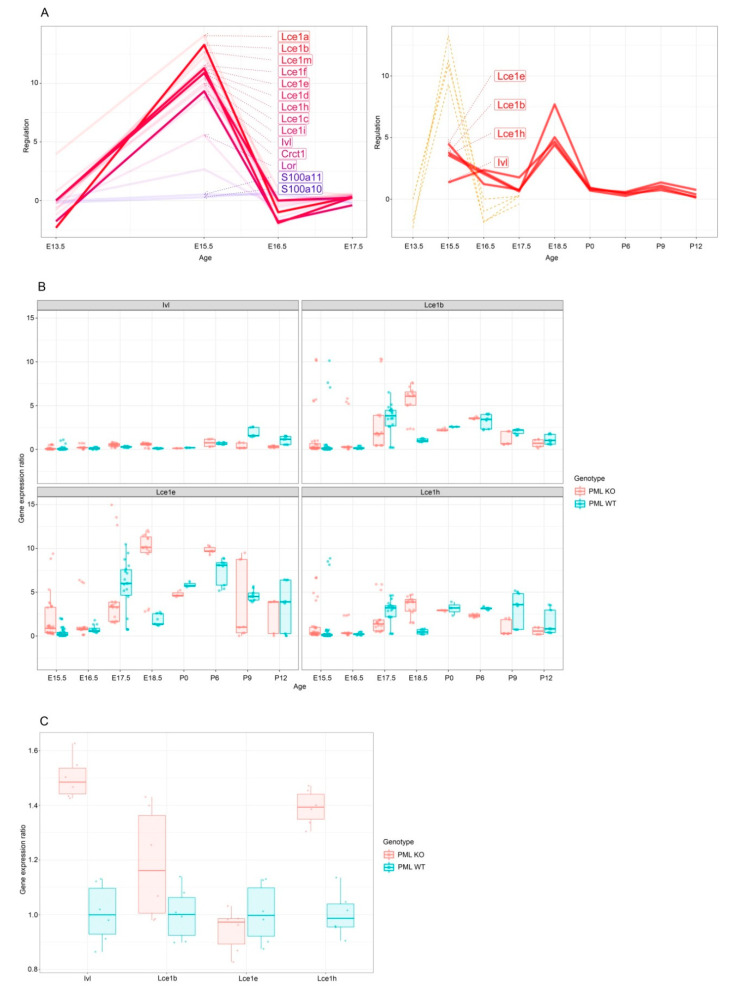
PML controls expression of the cornified envelope genes. (**A**) Graphs depicting differentially expressed cornified envelope genes. Left panel shows data from RNA-seq analysis. Right panel shows a comparison of data obtained from RNA-seq (stapled lines) and RT-qPCR (solid lines), respectively. (**B**) Box plots showing differential expression of selected genes from the cornified envelope region of the EDC gene locus by RT-qPCR analysis of PML-wt and PML-KO extracts. Differential expression at different time points during embryonic and postnatal development is shown. Each data point (dot) in box plot represents a single embryo. (**C**) Keratinocytes were isolated from the epidermis of PML-wt and PML-KO mouse embryos (stage E18.5) and subsequently passaged in culture. RNA was extracted from the cultured cells and selected genes from the cornified envelope region of the EDC gene locus were subjected to analysis by qPCR. Box plots show differential gene expression of selected genes. Each data point (dot) in box plot represents a single embryo.

**Figure 3 genes-11-01130-f003:**
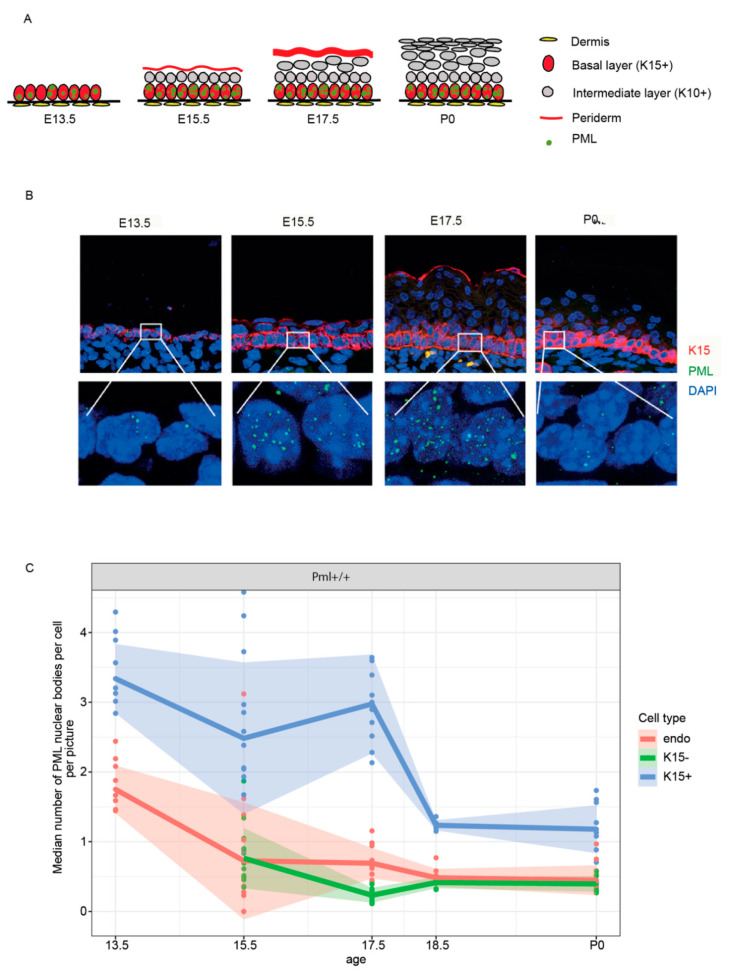
Elevated levels of PML bodies in basal epidermal cells during embryogenesis. (**A**) Schematic of epidermal cell layers during embryonic and postnatal development. Structural layers of epidermis at E13.5, E15.5, E17.5, and postnatal day (P) 0 is depicted. PML nuclear bodies are predominantly observed in the basal cell layer during embryogenesis. (**B**) Paraffin embedded sagittal sections obtained from developmental stage E13.5, E15.5, E17.5, and P0 were immunofluorescence (IF) labeled using antibodies against K15 (red) and PML NB (green). 4′,6-diamidino-2-phenylindole (DAPI) is shown in blue. The images are projections of multiple confocal Z-stacks. White rectangles highlight zoom-in of basal cells containing PML. (**C**) Quantitative assessment of PML bodies in different layers of epidermis obtained from E13.5, E15.5, E17.5, E18.5, and P0 mice. Each data point represents the average of PML bodies per cell from a single embryo.

**Figure 4 genes-11-01130-f004:**
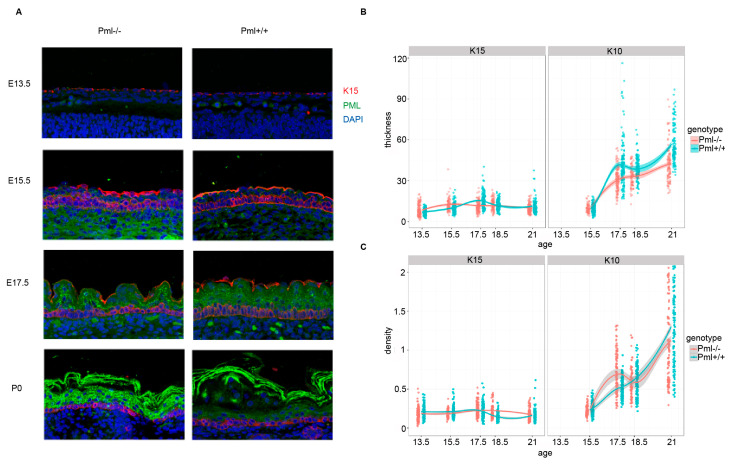
PML affects epidermal thickness during embryogenesis. (**A**) Representative sagittal images of epidermis derived from PML-wt and PML-KO E13.5, E15.5, E17.5, and P0 mice. Sections were IF-labeled using antibodies against K15 (red) and K10 (green). DAPI is shown in blue. The images represent projections of multiple confocal Z-stacks. (**B**) Quantification of epidermal thickness. The plotted lines represent average thickness (±STD) of K15 and K10-positive epidermal layers. Each data point represents measurements from a single microscopic field. More than 50 microscopic fields from 5–7 embryos/newborn mice per developmental stage were analyzed. (**C**) Quantification of cell density in basal and spinous layer of epidermis at different stages of embryonic development. Data are represented as aforementioned in (**B**).

**Figure 5 genes-11-01130-f005:**
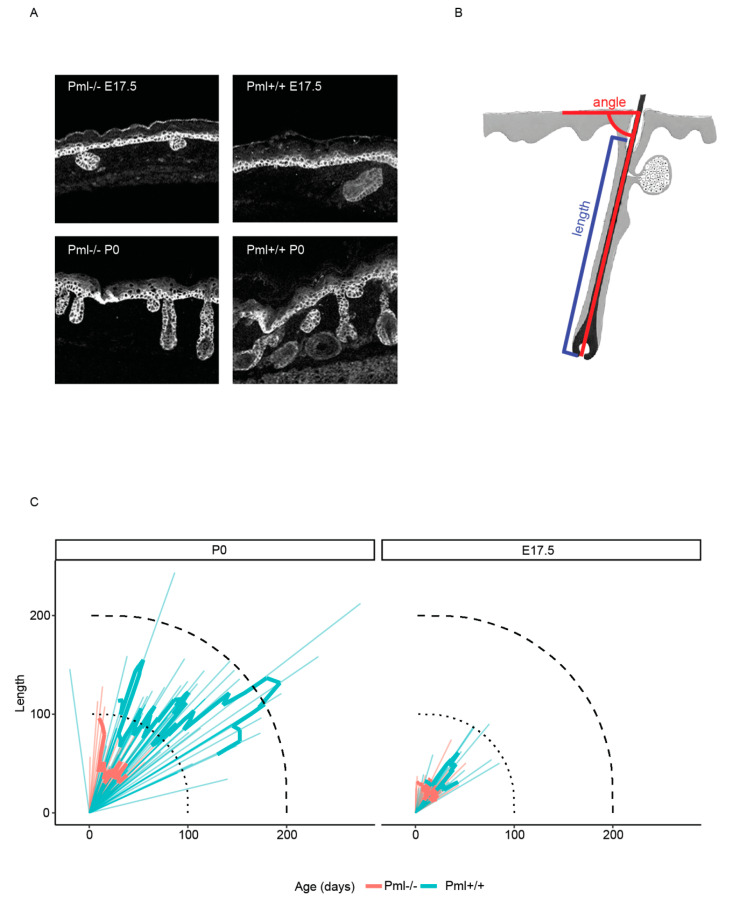
PML affects hair follicle size and orientation. (**A**) Representative sagittal images showing the morphology of hair follicles from PML-wt and PML-KO mice at stage E17.5 and P0. Samples have been IF labeled using antibodies against the basal keratinocyte marker K5 (red). (**B**) Schematic showing method for calculation of hair follicle size and orientation relative to the skin surface. (**C**) Polar diagram showing length and relative orientation of hair follicles from *Pml*+/+ and *Pml*−/− mice at the P17 and P0 developmental stages of mouse development.

**Table 1 genes-11-01130-t001:** qPCR primers.

Gene	Forward Primer	Revers Primer
*Ivl2*	TGGTTCAGGGACCTCTCAAG	ATGTTTGGGAAAGCCCTTCT
*Lcs1h*	CCTGCTGTAGCTTGGGTTCT	GAGATCTGTGGGGTCTGTGG
*Lce1e*	CAAATGCCAGATCCCAAAGT	AACCCAAGCTACAGCAGGAA
*Lce1b*	GTTGTTGTGGCTCCAGCTCT	CACCACTGCTACAGCATCCA

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
