# Peer review of "PML Regulates the Epidermal Differentiation Complex and Skin Morphogenesis during Mouse Embryogenesis"

_genes, 2020, doi:10.3390/genes11101130_

Round 1
Reviewer 1 Report
In this study, the authors examine the role of the promyelocytic leukemia (PML) protein in the epidermal development of embryonic mice using RNA-seq, and immunohistological approaches in PML wild type (WT) and knockout (KO) animals. The PML protein forms nuclear bodies and these subdomains are likely regulators of gene expression at certain loci either by physically interacting with these loci, providing platforms for post-translational modification of transcription factors and/or epigenetic modification of the chromatin in the vicinity of these bodies.
The authors find that PML loss appears to dysregulate gene expression in the epidermal differentiation complex (EDC), a locus that contains a large number of genes involved in epidermal development, which phenotypically correlates with delayed epidermal stratification and impaired hair follicle growth. The study will be a good edition to the canon of literature showing various developmental roles for PML. However, it falls short in demonstrating if PML bodies are implicated in regulation of the EDC or not, and there are a few other minor issues that can be addressed by further discussion and text edits.
Major Comment
1) The authors should show the localization of the EDC locus versus PML NBs by immunolocalization of PML protein plus DNA fluorescent in situ hybridization (FISH) and/or RNA-FISH for individual genes in the EDC locus. This data would really complement the RNA-seq data and clearly implicate PML bodies (rather than a downstream gene regulated by PML) in the regulation of this multi-gene locus. This is a fairly quick experiment, in particular for RNA-FISH. NOTE: positive localization of the EDC genes at a PML body is not a pre-requisite for publication but it does greatly change the interpretation of the data, and the shape of future experimentation. Thus, this experiment has to be done.
Minor Comments.
1) In the Introduction, there are several manuscript references that should be used instead of the BioGrid web URL to support the statement about the large number of PML-associated proteins. These references include the actual reference for the automated BioGrid Database (Oughtred et al., 2019 PMID: 30476227), the Nuclear Protein Database, which is human annotated database of nuclear protein interactions/localizations (Dellaire et al., 2003 PMID: 12520015) and work by Liu et al., 2010 PMID: 20823370 describing a large number of PML-interacting proteins.
2) Also in the Introduction, the authors should consider describing more recent examples of active genes localized at PML bodies, in addition to the often quoted MHC II locus, such as the DDIT4/REDD1 gene (Salsman et al., 2017 PMID: 28332630); noteable for the use of RNA-FISH showing specifically that the transcribed locus is localized at the PML body.
3) Although the Discussion talks about the broader role of PML in development in a brief summary statement, the Introduction would also benefit from a deeper overview of the PML’s role in development, including aberrant developmental programs such as leukemia initiating cells (LICs)(Ito et al., 2008 PMID: 18469801). The reasons for looking at the epidermis are also not really clear in the Introduction material. Can the authors explain briefly in more detail why they looked at the epidermis in the first place, e.g. at the beginning of the Results section perhaps? Was it from EDC gene expression in preliminary RNA-seq data, perhaps derived from embryonic fibroblasts from PML WT and KO mice?
4) None of the key transcription factors and developmental programs involved in epidermal development are discussed in the Discussion. Building on the Major Comment above, are PML bodies involved in EDC dysregulation? Or is the impact indirect, where PML loss results in alterations in the post-translational modification (or gene expression) of one or more transcription factors or chromatin regulators that are known to interact with the EDC? Some literature supported speculation is required here in addition to direct experimental evidence of PML body interaction with the EDC locus (or NOT).
Reviewer 2 Report
In this manuscript, Polec et al. study the function of the promyelocytic leukemia protein in regulating epidermal differentiation and morphogenesis in the mouse. By perfoming gene expression analysis studies in the embryonal epidermis of wt and Pml-/-mice, they found that genes belonging toa large gene cluster that contains epidermal differentiation genes are deregulated in the absence of Pml. Accordingly, morphological features of epidermis and hair follicles are altered in Pml-/- mice.
This is an interesting article that confirms the role of PML in regulating tissue development and morphogenesis. Unfortunately, it is hard to understand what the phenotype is, given that there are few inconsistencies in the description of the experiments. Mainly, in Figure 2B and C, contrary to what the authors assert, it appears that LCE genes are more expressed in wt embryos. There must be a mistake in labeling. Similarly, in Figure 4 it appears that wt epidermis is thicker than Pml-/-, contrary to what asserted in the text, which however is yet differently presented the discussion.
Besides these two major points, pathway enrichment in Figure 2A is quite difficult to comprehend. Gene names are very small. In the left hand panel only few genes are indicated. The comparison between Q-PCR and RNA seq shown in the right panel displays different time points and the gene names are again too small. Most importantly, it is not clear that one confirms results from the other experiment as the time points are different.
Minor comments
- In the introduction (lines 47-48) the authors state that the PML-NBs interact with the MHC locus in a manner that is regulated by interferon-induced gene regulation, however Shiels et al 2001 demonstrate that the interaction of PML and the MHC locus is independent of interferon gamma treatment.
Round 2
Reviewer 1 Report
Reviewers concerns were addressed. For immuno-FISH experiments, it is very much appreciated that experimental difficulties have made this hard to address in a timely manner.